# Frequency- and Temperature-Dependent Uncertainties in Hysteresis Measurements of a 3D-Printed FeSi wt6.5% Material

**DOI:** 10.3390/s24092738

**Published:** 2024-04-25

**Authors:** Bence Kocsis, Tamás Orosz

**Affiliations:** 1Department of Material Science, Széchenyi István University, 9026 Győr, Hungary; kocsis.bence@ga.sze.hu; 2Department of Power Electronics and Electric Drives, Széchenyi István University, 9026 Győr, Hungary

**Keywords:** additive manufacturing, uncertainties, FeSi steels, magnetic hysteresis

## Abstract

Additive manufacturing of soft magnetic materials is a promising technology for creating topologically optimized electrical machines. High-performance electrical machines can be made from high-silicon-content FeSi alloys. Fe-6.5wt%Si material has exceptional magnetic properties; however, manufacturing this steel with the classical cold rolling methodology is not possible due to the brittleness of this material. Laser powder bed fusion technology (L-PBF) offers a solution to this problem. Finding the optimal printing parameters is a challenging task. Nevertheless, it is crucial to resolve the brittleness of the created materials so they can be used in commercial applications. The temperature dependence of magnetic hysteresis properties of Fe-6.5wt%Si materials is presented in this paper. The magnetic hysteresis properties were examined from 20 °C to 120 °C. The hysteresis measurements were made by a precision current generator–based hysteresis measurement tool, which uses fast Fourier transformation–based filtering techniques to increase the accuracy of the measurements. The details of the applied scalar hysteresis sensor and the measurement uncertainties were discussed first in the paper; then, three characteristic points of the static hysteresis curve of the ten L-PBF-manufactured identical toroidal cores were investigated and compared at different temperatures. These measurements show that, despite the volumetric ratio of the porosities being below 0.5%, the mean crack length in the samples is not significant for the examined samples. These small defects can cause a significant 5% decrement in some characteristic values of the examined hysteresis curve.

## 1. Introduction

Electrical machines are indispensable in energy generation, manufacturing, and e-mobility applications. Many different electrical machine design methodologies have evolved during the last century, and many different objectives have been considered during the design and manufacturing process. An essential purpose during the design and development is to increase the efficiency of electrical machines, which is the goal of the current EU directives, to produce electrical machines of IE4 or better efficiency class [1,2,3]. Another aim is the introduction of the circular economy aspects into the design and manufacturing of electrical machines. To achieve this goal, different, more complex objectives (e.g., total cost of ownership, using recyclable or remanufactured components), design and optimization methods should be applied for electrical machine development, which can consider not only the losses but also the environmental and economic cost of the machine [4,5,6,7].

Additive manufacturing (AM) is a promising technology that can help create novel designs and rapid prototyping. The gas-atomized metallic powder used for printing soft magnetic parts can be produced from recycled materials, which can increase the recyclability rate of electrical machines and decrease the carbon dioxide footprint of raw material production at the same time [8,9,10]. Printing electrical conductors is the most mature 3D metal printing technology, and the performance of additively manufactured parts can reach conventional conductors due to the applied technology, which can print unconventional shapes with high precision [3,11]. Printing the machines’ hard and soft magnetic steel–based parts is still challenging, which is an actively researched area with a lower technical readiness level [3,12].

There are many soft magnetic alloys (iron–cobalt (FeCo), iron–nickel (FeNi), and iron–silicon (FeSi) [13,14,15,16,17]) used to create rotors and stators for electrical machines. FeCo alloys provide the highest saturation magnetization of these alloys, while FeNi alloys have much higher maximum relative permeability than FeCo alloys. Both of these alloys have good 3D printing properties. However, they are costly. Considering the price and the performance of the applications, FeSi-based materials with 2–7% silicon content guarantee an excellent choice [18]. Most 3D electrical machine printing papers focus on this material. Additionally, its excellent performance per cost ratio, high magnetic saturation, high maximum relative permeability, and low intrinsic coercivity can be achieved. The appropriately designed FeSi-material-based stator and rotor materials can have low hysteresis and eddy current loss parameters up to kHz excitation frequencies [15,19,20].

It is well known and widely researched that FeSi steel has 6.5 wt% Si, which offers the best soft magnetic proprieties for the magnetic circuit design [18]. However, due to their brittleness, the manufacturing and applicability of silicon steels with 4 wt% or greater silicon content has become challenging, and it is not possible to produce thin sheets through the conventional rolling process [21]. These silicon steels are usually manufactured by using special technology, like rapid quenching or chemical vapor deposition techniques [18,22,23,24]. It was shown in 2016 [25] that additive manufacturing can be a possible technology to create FeSi steels with high silicon content. Since that time, many researchers have investigated the applicability of 3D-printed FeSi materials with silicon content in this range [3,18,19,20,24,26]. Powder bed fusion of metals with a laser beam (PBF) is one of the most widely used 3D printing technologies, which can be used for processing these materials. This technique refers to metallurgy methods, like direct metal laser sintering, electron beam melting, and selective heat sintering, as well as selective laser melting and selective laser sintering [27,28,29,30]. The difficulty of printing high silicon steels is due to the brittleness caused by the high silicon content, primarily due to the $B_{2}$ and $DO_{3}$ ordered phases and also due to the internal stresses caused by repeated thermal cycling. The latter effect can be reduced by increasing the heating of the table, i.e., by reducing the temperature gradient. The reduction in the amount of ordered phases can be achieved by rapid cooling, i.e., cooling of the table. Eliminating the two leading causes of cooling down requires opposite interventions during the printing process [31].

Stornelli et al. compared the magnetic properties of Fe-6.5wt%Si alloys with the properties of Fe-3wt%Si alloys [24]. The compared materials were printed using selective laser melting technology. The parameters were optimized using 20–20 samples for both materials. It was proved by the measurements that the Fe-6.5wt%Si materials have 50% less magnetic losses compared with the printed Fe-3wt%Si materials. However, its performance is still far from the laminated Fe-3wt%Si-based materials. Structurally layered Fe-6.5wt%Si material was presented in [32], where this material can achieve shallow core loss. It was 52.5 W/kg at 1 kHz, while creating 0.2 mm thick laminations was possible. Another paper examined the effect of particle size distribution on the magnetic properties of Fe-6.5 wt% Si powder cores [33]. It was concluded that the particle size significantly impacts the core losses. It was found that there was a significant 53% difference between the samples with the biggest and the smallest particle size. Other authors [34,35] tried to dope the Fe-6.5 wt%Si powders with Ti and Co to improve the material’s internal microstructure to reduce the material’s coercivity. Another research direction was to increase the pressure during the material manufacturing to make more compact electromagnetic cores [36].

In this article, the temperature dependence of the magnetic hysteresis of a Fe-6.5 wt% Si powder core was examined. The main difference between this research and the previously proposed research processes is that, in this research, ten toroidal samples were produced at once. They were examined using the same methods, computer tomography and optical microscopy, for structural defects and statical magnetic hysteresis measurements in a heat chamber. Besides the temperature dependence of the samples, we would like to examine the effect of the manufacturing defects on the magnetic hysteresis material.

## 2. Materials and Methods

### 2.1. Sample Preparation

A powder bed fusion technology made the printing process with an EOSINT M270 machine (EOS GmbH, Offenbach am Main, Germany). This equipment has an ytterbium fiber laser with 200-watt nominal power. It has a 100 μm nominal diameter with a Gaussian power distribution curve. The platform temperature was kept at 100 °C during the building process with a nitrogen shielding gas. Ten toroidal samples were printed in the same, which has the same dimensions: d=7 mm inner diameter, D=12 mm outer diameter, and h=3.5 mm height.

The optimal volumetric energy density (VED) value described above is obtained in a previous research paper by the following formula [29]:(1)$$VED=\frac{P_{laser}}{v_{scan}\cdot h\cdot d}.$$
where *P* is the applied power of the laser (*W*). $v_{scan}$ is the laser scanning speed in (mm/s). *h* is the hatch width [μm], while *d* is the layer thickness (μm). During the optimization process, $P_{laser}=165$ W and $v_{scan}=700$ mm/s parameters were derived [30]. These parameters are close to other authors’ settings [18], which are used in similar equipment.

The Fe-6.5wt%Si powder for the PBF-L-based sample preparation process was manufactured by Changsa TIJO Metal Material Co., Ltd., (Changsha, China) using a gas atomization process. During the preparation phase, laser diffraction and scanning electron microscopy (SEM)–based measurements were performed to check and ensure the powder quality according to the GB/T 19077-2016 standards [37]. The resulting image of the SEM, with the particles’ shape and morphology, can be seen in Figure 1, which follows the desired spherical shape with a smooth and clean surface, besides the required chemical distribution, which is necessary for printing Fe-6.5wt%Si alloy.

A SEM-energy dispersive spectroscopy (EDS) was used to measure the chemical composition of the powder particles, and a laser diffraction method was used to determine the particle size distribution of the used powder. The results of these measurements are plotted in Figure 2a,b. Their chemical composition is close to the desired Fe-6.5wt%Si alloy. Their characteristic size falls within the size range the laser diffraction study gave. The mean particle size was given as 33 μm, while most of the particles (95%) were characterized as having a particle size greater than 20 μm and smaller than 55 μm, according to the GB/T 19077-2016 and ISO 13320-2020 standards [37,38].

### 2.2. Hysteresis Measurements

The applied computer-controlled measurement system was initially designed to characterize the static hysteresis model of the ferromagnetic steel sheets for the Preisach-model-based characterization and modeling of magnetic steel sheets [39,40,41]. Well-known materials and their characteristics were used to validate the measurement system, which was successfully used in many projects to measure the effect of the manufacturing process on the characterization of different soft magnetic steel sheets [42].

The main working principle of the applied system is depicted in Figure 3. During the measurements, two coils were wound around the printed samples, where the primary coil consists of 30 turns ($N_{1}$), while the secondary has only 20 turns ($N_{2}$) due to the small size of the toroidal sample; it was not possible to increase the number of turns if we wanted to use a copper wire that can withstand relatively high currents with adequate cross section.

The primary coil was excited by a computer-controlled current generator, while its current value was measured on a precision resistance by an NI PCI-6251 data acquisition card with a 16-bit resolution. The secondary coil was realized as an open circuit during the measurement. Both the current of the primary circuit and the voltage of the secondary circuit were sampled simultaneously. The noise of the measured signals was filtered by an FFT filter, which was realized in the card’s LabView -controlled measurement software. The magnetic field strength can be calculated from the current value of the current using the following equation:(2)$$H=\frac{N_{1}}{l_{eff}}i_{1}\left(t\right).$$
where $i_{1}$ represents the current value in Amps, $l_{eff}$ is the mean length of the ring-shaped specimen, while *H* represents the magnetic field strength [40]. The magnetic flux density is calculated from the time function of the measured voltage ($u_{2}$) in the open circuit from the following formula:(3)$$B=\frac{1}{N_{2}A}\int_{0}^{t}u_{2}\left(\tau \right)d\tau .$$
where $N_{2}$ is the number of turns in the secondary coil, and *A* is the cross-sectional area of the toroid sample.

## 3. Results and Discussion

### 3.1. Metallurgical Properties of the Toroidal Samples

With the previously tested printing parameters, all samples were printed simultaneously without problems [29]. Optical microscopy was used to look for recurring phenomena of characteristic microstructural defects in several samples. A multifunctional Zeiss Axio Imager M1 microscope and AxioVision 4.8 evaluation software were used. X-ray diffraction was also performed using a Bruker D8 diffractometer to investigate the crystal structure of the laser-sintered samples. The mechanical and magnetic properties of additively manufactured parts are significantly degraded by cracks and porosities in the structure. Such defects do not comply with industry standards and hinder the widespread application of AM technologies. The keyhole pore is one of the most significant and common defects (Figure 4). Typical characteristics are that they occur at typical locations at the boundaries of the melt pool and have circular facies, as can be seen in Figure 4, where the marked keyholes are of the types described by S. Mohammad et al. [43]. In this image, the layers were built on top of each other from the upper-right corner to the lower left. Type I and type III are the result of depression zone fluctuation and gas expansion or vaporization. These types can be inferred from the near-circular pores at the bottom of the melt pool (type I) and at the boundary of the melting boundary (type III). Figure 4, marked in black, also clearly shows that the crack extends from porosity to porosity and is perpendicular to the direction of construction (z−axis). Cracks of this type are not formed due to inadequate fusion but typically caused by the relaxation of frozen internal stresses in this manner. Numerous studies have investigated the causes, locations, orientations, and prevention of porosity and crack formation [44]. In most cases, cracks parallel to the Z-axis (build direction) are caused by the combination of a high-temperature gradient perpendicular to the build direction and porosities (keyholes) that form between layers (somewhere in the melt pool). The temperature gradient is responsible for the internal stress, and the porosities act as stress concentration points where cracks can start or end—characteristic values of the examined hysteresis curve.

Figure 5 shows a cross-sectional view of one of the specimens parallel to the XY-scan plane, where the layers were formed pointing outwards from the plane of the image. The sample clearly shows the pattern of the 67° rotational laser scanning strategy and the nearly circular porosities at the interface.

The samples were examined by computer tomography (CT) to detect the microstructural defects (cracks, porosity) in the samples using a YXLON Y Modular system. Figure 6 shows the CT images from sample #1, sample #6, and sample #10. It can be seen from this image that sample #10 contains a significantly smaller number of cracks and porosity than sample #1 and sample #6. The difference between these samples is caused by many small uncertainties during the manufacturing of the samples. These uncertainties can come from different sources; for example, as previously seen, the size of the particles is not identical, there can be some local differences in the material composition between the different samples, the place of the sample in the table can have an effect, or the precision of the laser can affect the manufacturing process.

It can be seen from Table 1 that the difference between the measured volumes is negligible, less than 0.5%. The volume of the toroid sample is nearly the same at 145.5 mm^3^. The amounts of porosity and crack volumes in the material are not significant. Sample #6 has the highest porosity value, 0.5% of the material, while #10 has the fewest manufacturing defects; only 0.019 % of the total volume is intended. Ideally, the value of the porosity is less than 0.01%, while there is an inverse relationship between cumulative crack length and porosity. This phenomenon was first described by M. Garibaldi et al. for Fe-6.5wt%Si alloy [25]. It can be concluded that the porosity in the manufacturing samples was negligible, and the cumulative length of the cracks was insignificant, so a near-optimal laser sintering setup was used for the printing.

In Figure 7, an X-ray diffractogram shows that the second peak 2θ=77 is absent in this case, indicating anisotropy of the printed material; i.e., epitaxial crystal growth is observed in the direction of the build (z). Subsequent heat treatment can reduce this effect. Moreover, this figure agrees with the previously published image in [29], which shows that the structure of the material agrees with the structure of the previously experimentalized material. In this paper, we have not made a heat treatment on these samples; we would like to examine how these negligible manufacturing errors and deformities affect the magnetic hysteresis uncertainties.

### 3.2. Uncertainties in the Magnetic Hysteresis Measurement

Before examining the uncertainty of the temperature effect, the measurement system’s accuracy was tested by repeatedly measuring the same sample. Sample 1 was used for this measurement, which views before and after the winding can be seen in Figure 8a,b. Due to the small size of the sample, we had to use copper wires with relatively small cross sections to wind as many primary and secondary turns as possible. The applied current generator can produce 30 Amps; however, for the above 10 Amps, we found that the wires were warming rapidly, which can affect the measurement. Therefore, we decided to limit the exciting current to 10 Amps and the magnetizing force to 8000 A/m. That was the maximal curve that we examined the measurements, and the results of this setup are presented in this paper.

A 120 °C curve is shown in Figure 9 to show how the measured static hysteresis curves change during the function of the temperature. We have marked some characteristic points in the figure and on the reported hysteresis curve to characterize the effect of increasing temperature by changing these points. These points are the following: the coercive force ($H_{c}$), marked by the intersection of the hysteresis curve and the B=0 axis; the remanent magnetic field ($B_{r}$), marked by the intersection of the hysteresis curve and the H=0 axis; and $B_{p}$, which is the magnetic flux density value at 7000 A/m magnetization force.

Based on previous experience and calculations, the accuracy of the hysteresis measuring device is around 0.5%. In the first measurement, we wanted to check how the accuracy of the measurement evolves for the three characteristic values mentioned above. For this purpose, measurements on sample #1 were repeated ten times to filter out the unnecessary peaks caused by a switch-on. The measurement results are in Table 2. The measured data show that $B_{r}$ is the most sensitive to the measurements, with a relative value of about 1%, while $H_{c}$ is the most stable of the three selected characteristics, with a sensitivity of less than 0.2%. These numbers characterize the sensitivity of our measurements, all of the changes in the hysteresis curve parameters that are higher than the numbers mentioned above caused by the thermal and frequency dependency of the examined samples.

### 3.3. Temperature-Dependent Magnetic Hysteresis Measurement

The hysteresis curve was measured under the same conditions in all ten samples by applying different frequencies and temperatures. The exciting current was set to 8000 A/m during the measurements, while the frequency was set to 5 Hz, 50 Hz, and 200 Hz. The temperature was set by an Angeloni Tect DY110 SP type climatic chamber to the following values: 20 °C, 40 °C, 60 °C, 80 °C, 100 °C, and 120 °C.

Figure 10 shows the hysteresis curves at 20 °C at the different measured frequencies, 5 Hz, 50 Hz, and 200 Hz. It can be seen from the image that, in agreement with the expectations, the area of the hysteresis loop is increasing due to the increased frequency.

The previously shown Figure 9 illustrated the hysteresis curves at 20 °C and 120 °C in the case of sample #1. Both of these curves were measured at 200 Hz frequency. It can be seen from the image that $B_{r}$, $H_{c}$, and Bp transform with different extents. The value of $B_{r}$ decreases from 0.4986 T to 0.3944 T, which is a relatively significant change; this value is decreased by 20% during 100 °C due to the heating. The change in $B_{p}$ seems more significant in absolute values (Figure 9). However, it decreased from 1.317 to 1.2711, which is only a 3.5% change in the relative scale. The value of $H_{c}$ is increasing from −955.8 A/m to −814.78 A/m, which is a relevant 14% change in relative units. These measurements suggest that the value of $B_{r}$ is the most sensitive to the temperature increase, while the value of Bp is most insensitive. However, as we saw during the CT experiments (Figure 6), sample #1 contained significantly more porosities and some cracks, which grouped in two areas along the ring. This can affect the temperature-dependent behavior of these characteristic points. To answer this question, first, we can check how these values transformed in the case of sample 10, which has the most idealistic porosity and crack-free parameters. Second, we can compare these characteristic values at room temperature in the ten samples. Table 3 contains the measured values of the different samples at room temperature, at the three examined frequencies, while all of the measured hysteresis curves are plotted at room temperature and at a 200 Hz excitation frequency in Figure 11.

As expected, after the optical microscopy and CT investigations, sample #7 and sample #10 had the best properties in both of the three examined parameters, which were expected due to the exceptionally low porosity and crack-free parameters of these two samples. From the CT examination (Table 1), we have seen that the highest porosity value is less than 0.5% in the measured porosity values, while there is a 5% uncertainty in the $B_{r}$ values. This value seems the most sensitive to manufacturing errors, while $B_{p}$ is the most robust. The measured uncertainty is in the range of the measurement precision, which is around 1 %. Comparing the previously shown sample #1’s magnetic parameters with sample #10, the $B_{r}$ value is 0.588 T at room temperature, which decreases to 0.4478 T at 120 °C, which is a 23% decrement. The $H_{c}$ value changes from −1058 to −859, while $B_{p}$ changes from 1.382 to 1.31 in the case of this sample. The change in $H_{c}$ is around 18%, while the change in $B_{p}$ is around 3.7% in the values. These results show that the porosities can cause a significant difference in the measured material properties instead of the small number of cracks, mainly a 5% difference in the $B_{r}$ values. However, the trend of these values changing is similar in both examined samples. This can be an essential point for electrical machine designers who want to consider the temperature dependence of these materials. The selected magnetic hysteresis model should mimic this behavior.

Another conclusion of the results is that these uncertainties do not depend on the strongly applied frequency. It seems that $B_{r}$ has the same, around 5%, uncertainty in all measured frequencies. This supports the previous statement that these uncertainties depend mainly on the cracks and the porosities of the prepared samples. However, the bandwith of the measurements are very narrow, and the statements can be valid for rotational machines only; a wider bandwidth should be applied in a future research to check these dependencies on kHz frequencies. Table 4 and Figure 12 show the temperature dependence of the $B_{r}$ values at the examined frequencies. Meanwhile, the value of $B_{r}$ is monotonically decreasing as the temperature increases, and the rate of this decrement seems not affected by the frequency significantly, and in the case of Figure 13, the $H_{c}$ value has different behavior. The $H_{c}$ value increases by the temperature; however, it seems the increment rate is significantly higher at higher frequencies.

## 4. Conclusions

Additive manufacturing is a promising technology that can produce high-silicon-content silicon–iron alloys for electric machines. The exceptional magnetic behavior of these materials is well known; however, due to their brittleness, it is not possible to produce these materials with conventional technology. Powder bed fusion technology is one of the possible alternative technologies that can offer a solution to produce Fe-6.5 wt%Si steels for electrical machines. Some previous works published the possibility and difficulties of manufacturing this material; the first measurements on these materials also proved its low core loss and good magnetic properties. This work examined the temperature dependency of the magnetic hysteresis properties of this Fe-6.5 wt%Si material. The applied methodology differs from that of similar papers because we measured all of the produced samples at once, not only the ones that had the smallest number of manufacturing defects (cracks, porosity). Three parameters were used to characterize and measure these uncertainties in the different shapes of B-H curves: the coercive force ($H_{c}$), where the B value is zero; the remanent magnetic flux ($B_{r}$), where the H value is zero; and the peak value of the flux, which is calculated at 7000 A/m ($B_{p}$). It was found that these values depend differently on the temperature increments. The $B_{r}$ value is the most sensitive to the manufacturing uncertainties, while the value of $B_{p}$ is the more insensitive parameter from this point of view. While the worst manufactured sample contains less than 0.5% porosities and cracks, its $B_{r}$ value decreases by more than 5% compared with the best-manufactured sample. Another result of the measurements is that the temperature dependency of these characteristic values seems to be independent of the ratio defects. In further work, we would like to examine the effect of the heat treatment on the parameters and these uncertainties. It would be interesting to examine whether applying an appropriate heat treatment can decrease this 5% difference in the $B_{r}$ characteristic parameter.

## Figures and Tables

**Figure 1 sensors-24-02738-f001:**
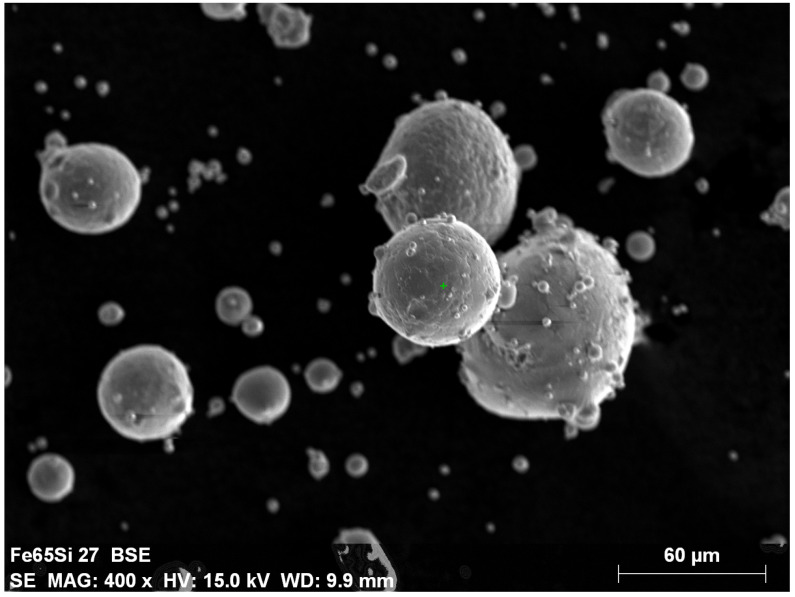
Scanning electron microscopy (SEM) image of the applied Fe-6.5wt%Si powder.

**Figure 2 sensors-24-02738-f002:**
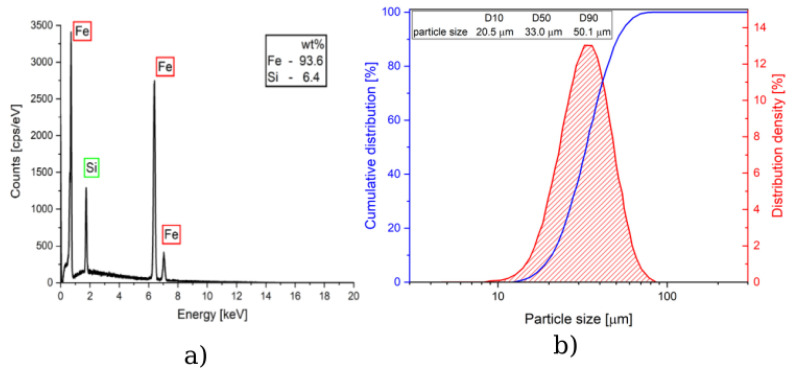
(**a**) Analysis of the chemical composition of the Fe-6.5wt%Si used by SEM–EDS, while the (**b**) part of the image shows the particle size distribution of the used Fe-6.5wt%Si powder by the laser diffraction method.

**Figure 3 sensors-24-02738-f003:**
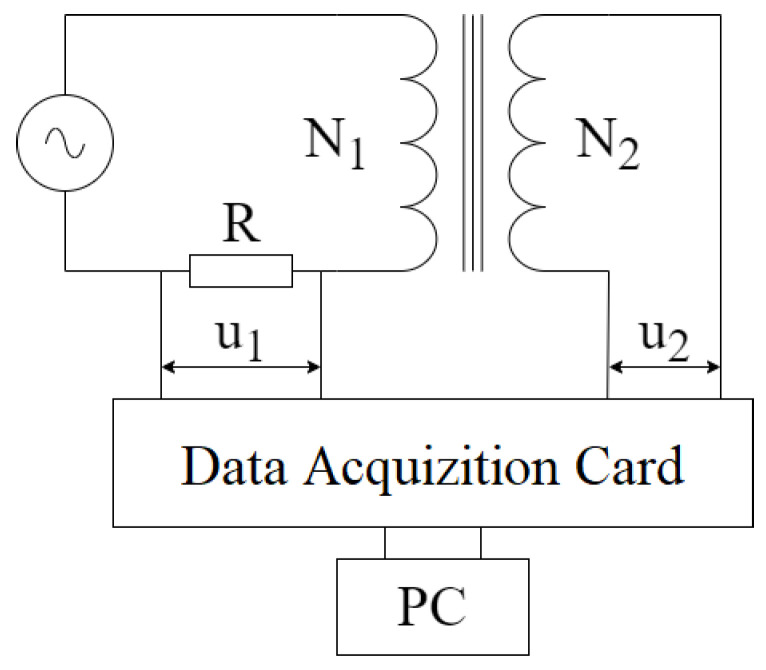
The main working principle of the computer-controlled measurement system consists of a computer-assisted current source, a primary and secondary coil around the specimen, and a data acquisition card [14].

**Figure 4 sensors-24-02738-f004:**
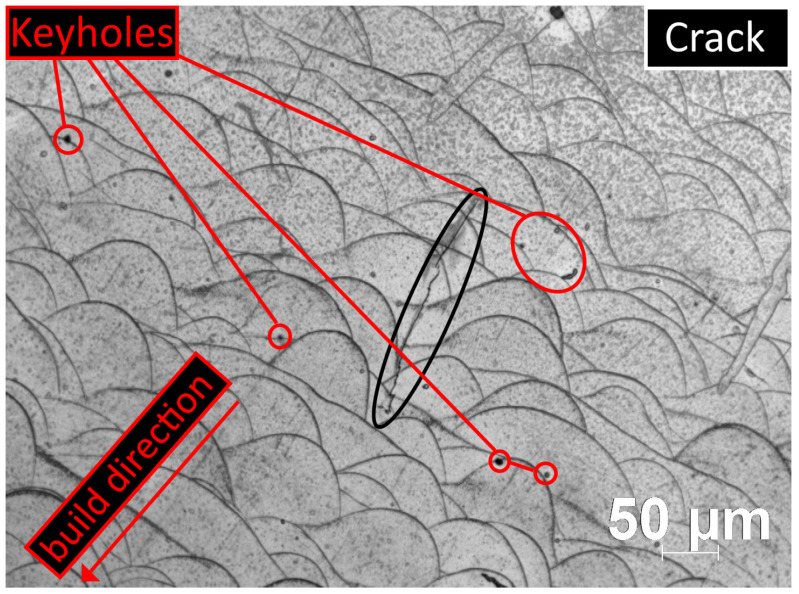
Optical microscope image from a 3D-printed toroidal sample perpendicular to the printing plane at 20× optical magnification. The keyholes and a crack are marked in the image.

**Figure 5 sensors-24-02738-f005:**
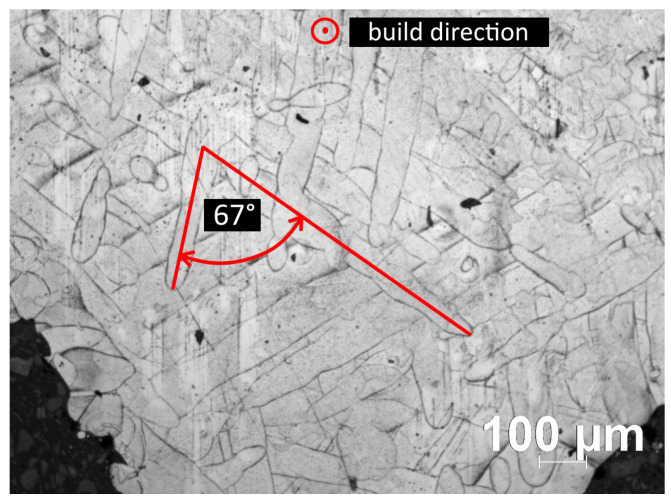
Optical microscope image parallel to the printing plane at 10× optical magnification.

**Figure 6 sensors-24-02738-f006:**
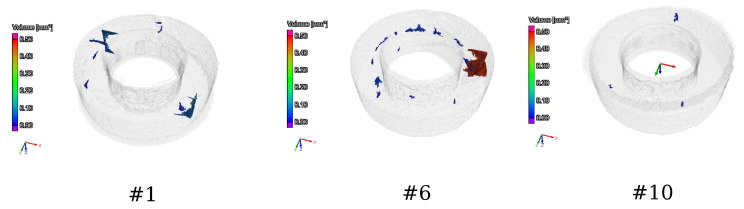
Computer tomography images from sample #1, sample #6, and sample #10 to illustrate and visually compare the distribution of the cracks in the samples.

**Figure 7 sensors-24-02738-f007:**
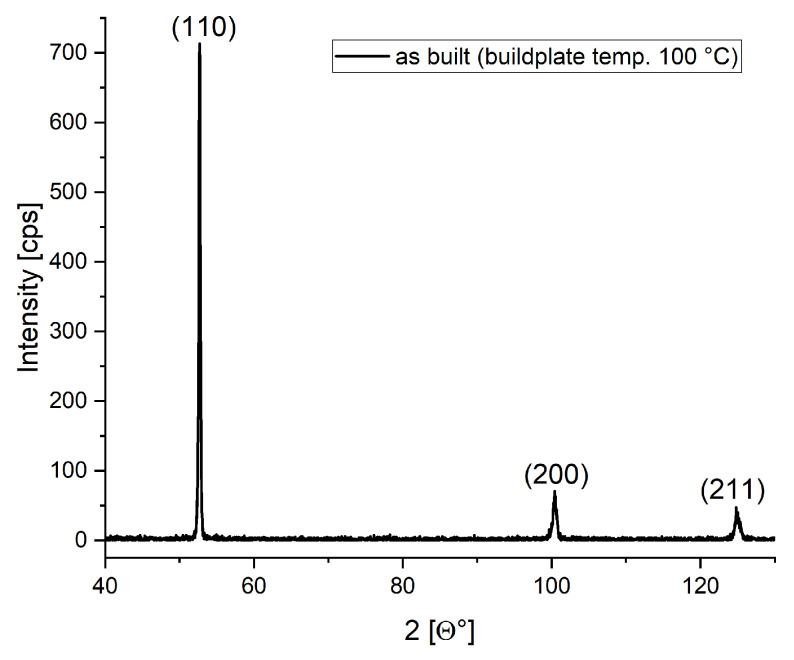
X-ray diffractogram from the as-built state of a toroidal sample.

**Figure 8 sensors-24-02738-f008:**
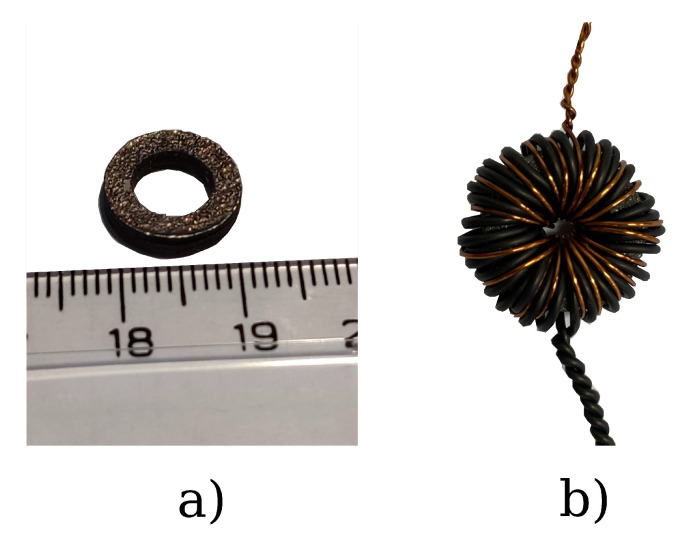
Image (**a**) shows the measured toroidal sample (sample #1), while image (**b**) shows the measurement setup with 30 primary and 20 secondary turns.

**Figure 9 sensors-24-02738-f009:**
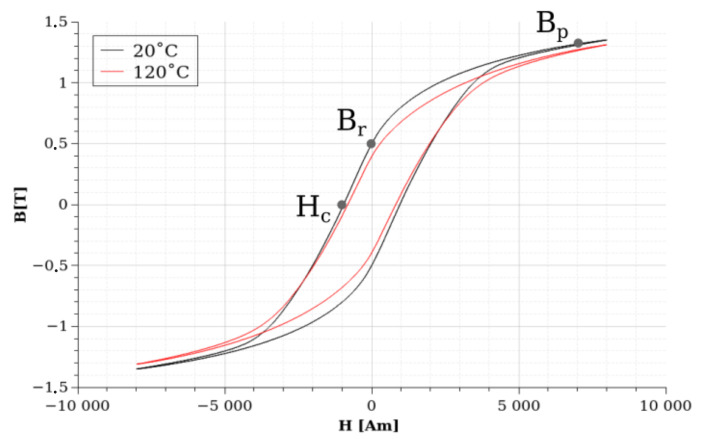
Frequency dependence of the examined FeSi materials’ magnetic hysteresis curve. The sample was measured at three different frequencies: 5 Hz, 50 Hz, and 200 Hz.

**Figure 10 sensors-24-02738-f010:**
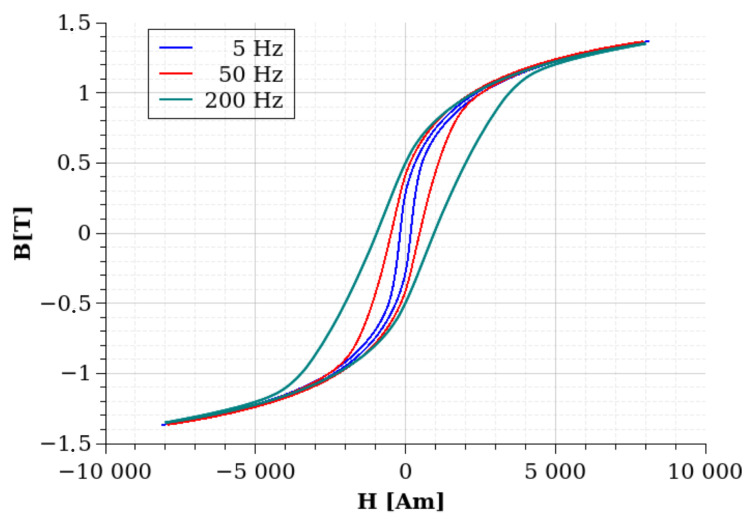
The measured hysteresis curves of sample #1 at the three examined frequencies: 5 Hz, 50 Hz, and 200 Hz.

**Figure 11 sensors-24-02738-f011:**
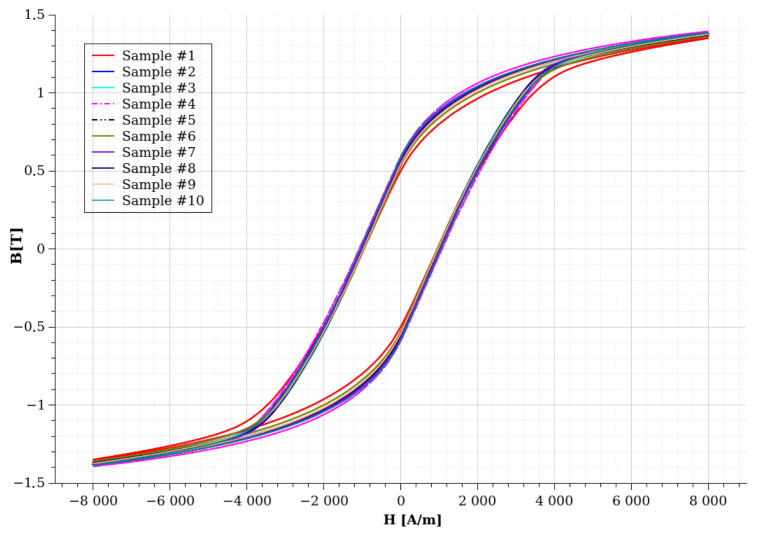
The measured hysteresis curves on all of the examined samples at 200 Hz.

**Figure 12 sensors-24-02738-f012:**
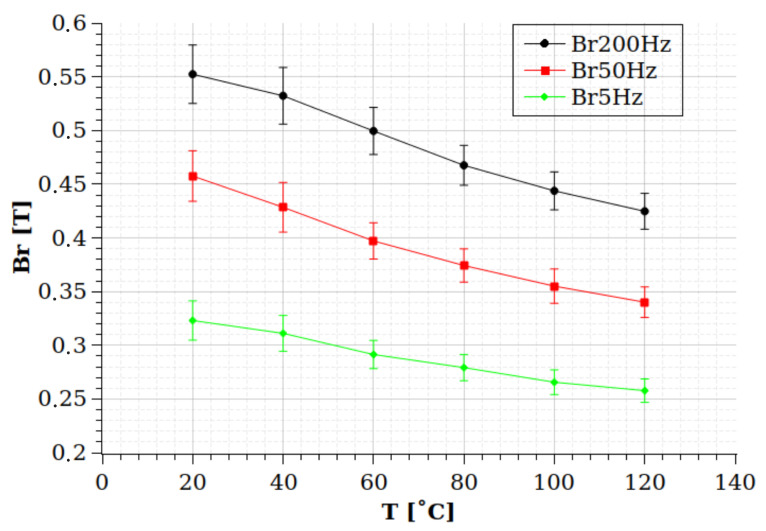
The dependence of the $B_{r}$ value on the temperature at 5 Hz, 50 Hz, and 200 Hz frequencies.

**Figure 13 sensors-24-02738-f013:**
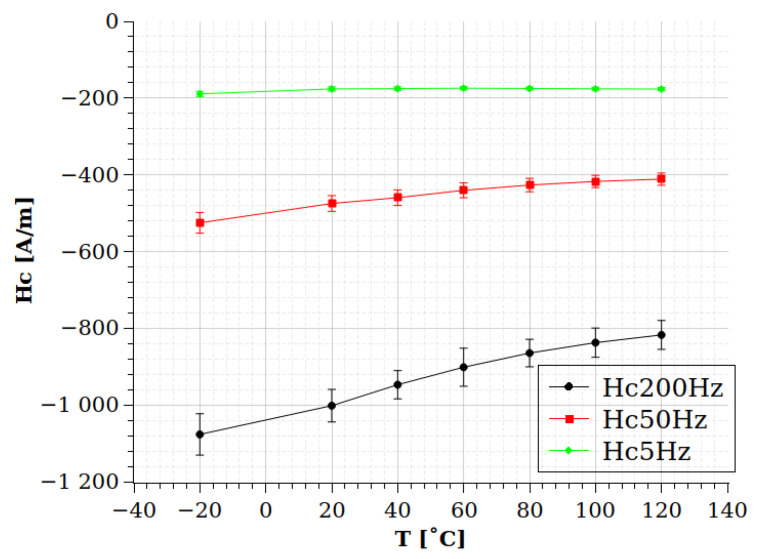
The dependence of the $H_{c}$ value on the temperature at 5 Hz, 50 Hz, and 200 Hz frequencies.

**Table 1 sensors-24-02738-t001:** Porosity values obtained based on CT reconstruction, in absolute and relative units.

Sample 1	Volume	Voids	Voids
#	mm^3^	%	mm^3^
1	145.718	0.286	0.196
2	145.542	0.54	0.369
3	145.71	0.358	0.245
4	145.448	0.545	0.374
5	145.614	0.409	0.28
6	145.387	0.667	0.456
7	145.466	0.607	0.416
8	145.608	0.426	0.292
9	145.516	0.546	0.374
10	146.01	0.028	0.019

**Table 2 sensors-24-02738-t002:** Results of the sample #1 measurements.

Measurement	Br [T]	Hc	B @ (7000 A/m)
#	[T]	[A/m]	[T]
1	0.53206	−1200.115	1.222023
2	0.53143	−1204.986	1.221007
3	0.53160	−1204.5206	1.220496
4	0.52968	−1197.1534	1.219436
5	0.52768	−1202.241	1.218853
6	0.52647	−1199.301	1.217969
7	0.52326	−1191.111	1.217387
8	0.51908	−1192.726	1.216071
9	0.51958	−1185.127	1.216902
10	0.51948	−1186.076	1.216210
Average	0.526	−1196.336	1.2186
STD	0.0053	7.234	0.0021
STD (%)	1.0085	0.6051	0.17

**Table 3 sensors-24-02738-t003:** Values of the characteristic points ($B_{r}$, $B_{p}$, and $H_{c}$) at 20 °C at different measurement frequencies.

	5 Hz			50 Hz			200 Hz		
Sample	$B_{r}$	$H_{c}$	$B_{p}$	$B_{r}$	H_c_	$B_{p}$	$B_{r}$	H_c_	$B_{p}$
	[T]	[A/m]	[T]	[T]	[A/m]	[T]	[T]	[A/m]	[T]
1	0.283	−176.133	1.327	0.413	−472.762	1.333	0.499	−955.790	1.317
2	0.316	−181.194	1.356	0.456	−493.116	1.361	0.564	−1019.370	1.345
3	0.326	−170.054	1.374	0.449	−448.451	1.376	0.540	−976.556	1.341
4	0.338	−187.390	1.375	0.473	−508.075	1.378	0.591	−1082.380	1.364
5	0.315	−172.265	1.350	0.470	−483.346	1.375	0.556	−996.580	1.351
6	0.314	−172.265	1.351	0.433	−451.613	1.352	0.531	−956.730	1.332
7	0.341	−175.818	1.380	0.479	−470.481	1.384	0.565	−1018.200	1.348
8	0.331	−171.990	1.373	0.465	−455.709	1.376	0.548	−978.860	1.342
9	0.319	−173.194	1.365	0.445	−465.794	1.367	0.543	−976.550	1.346
10	0.348	−181.408	1.378	0.492	−498.991	1.382	0.588	−1058.100	1.354
AVG	0.323	−176.171	1.363	0.458	−474.834	1.368	0.552	−1 001.912	1.344
STD	0.018	5.507	0.017	0.024	20.551	0.016	0.027	42.449	0.013
STD (%)	5.668	−3.126	1.235	5.142	−4.328	1.157	4.925	−4.237	0.949

**Table 4 sensors-24-02738-t004:** Temperature dependencies of $B_{r}$ values and their standard deviation at the different examined frequencies.

T	$B_{r}$ 200 Hz	$\Delta B_{r}$ – 200 Hz	$B_{r}$ – 50 Hz	$\Delta B_{r}$ – 50 Hz	$B_{r}$ – 5 Hz	$\Delta B_{r}$ – 5 Hz
[°C]	[T]	[T]	[T]	[T]	[T]	[T]
20	0.552	0.027	0.458	0.024	0.323	0.018
40	0.532	0.026	0.428	0.023	0.311	0.017
60	0.500	0.022	0.397	0.017	0.291	0.013
80	0.468	0.019	0.374	0.016	0.279	0.012
100	0.444	0.018	0.355	0.016	0.266	0.012
120	0.425	0.017	0.340	0.014	0.258	0.011

## Data Availability

Data are contained within the article.

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
