# Peer review of "Frequency- and Temperature-Dependent Uncertainties in Hysteresis Measurements of a 3D-Printed FeSi wt6.5% Material"

_sensors, 2024, doi:10.3390/s24092738_

Round 1

Reviewer 1 Report

Comments and Suggestions for Authors

This paper investigates the temperature dependence of magnetic hysteresis properties in 3D printed FeSi wt6.5% material. This material is promising for electrical machines due to its exceptional magnetic properties, but its brittleness makes it challenging to manufacture with traditional methods. Although the article contains interesting research, the article needs to be significantly improved.

- it is necessary to check the grammar of the text, since in some places it is not entirely clear what the authors mean, incorrect words are used somewhere.
- There are many places in the text where a new sentence does not begin after a period. Also somewhere a period is used instead of a comma. Lines 38, 43, 84-90, 97, 104, 109, 123 and others. Please check.
- Line 97 “Ten toroidal samples were printed in the same” explain what “in the same” means.
- After introducing the abbreviation SEM on line 109, the abbreviation must be used in the future (for example, lines 111, 115). The opposite situation is with EDS - the abbreviation is used without being introduced (must be added on line 115).
- Fig. 4, 5 it is necessary to indicate the direction of growth of the part when printing.
- table 1 – the ratio Voids, mm3 / Volume mm3 gives porosity values in % almost two times less than in the Voids, % column.
- Figure 7 – indicating of reflexes is necessary
- Figure 8 – it is necessary to improve the image quality.
- there is no discussion in the text of the effect of porosity on hysteresis. Why was the area with porosity research cited? Is there a maximum permissible porosity value that the authors wanted to obtain?
- line 217 – circumstances must be replaced with another word.

Author Response

Dear Reviewer,

 Thank you for your comments and suggestions, it helped us to improve our paper. Please, see our answers below they are  highlighted by blue color.

-it is necessary to check the grammar of the text, since in some places it is not entirely clear what the authors mean, incorrect words are used somewhere.

Thank you for this suggestion. The grammar of the paper was checked again, and the types and wrong wording were corrected.

- There are many places in the text where a new sentence does not begin after a period. Also somewhere a period is used instead of a comma. Lines 38, 43, 84-90, 97, 104, 109, 123 and others. Please check.

Corrected.

- Line 97 “Ten toroidal samples were printed in the same” explain what “in the same” means.

The more detailed explanation added to the text:

Ten toroidal samples were printed using the optimized printing process, which details were published in [29] (Kocsis, et al, 3D printing parameters optimization for Fe-6.5 wt% Si. Journal of Magnetism,  and Magnetic Materials). The toroidal samples have the same dimensions: d=7mm inner diameter. D=12mm outer diameter and h=3.5 mm height.

- After introducing the abbreviation SEM on line 109, the abbreviation must be used in the future (for example, lines 111, 115). The opposite situation is with EDS - the abbreviation is not introduced (must be added on line 115).

Corrected.

Fig. 4, 5 it is necessary to indicate the direction of growth of the part when printing.

These figures changed and the growth direction indiced on the figures.

- table 1 – the ratio Voids, mm3 / Volume mm3 gives porosity values in % almost two times less than in the Voids, % column.

Thanks for checking the values; the quantities of the two tables were changed during the formatting. Corrected.

- Figure 7 – indicating of reflexes is necessary

Corrected.

- Figure 8 – it is necessary to improve the image quality.

Corrected

there is no discussion in the text of the effect of porosity on hysteresis. Why was the area with porosity research cited? Is there a maximum permissible porosity value that the authors wanted to obtain?

Some more explanation added to line 175-183 to describe this effect. However, the role of the paper is not to analyse the functional dependence of the hysteresis from the porosity ratio. This can be an interesting topic in the following paper to examine it in more details.  We tried to create similar toroids by applying the same instruction set on the same machine; however, in this case, uncertainties can cause cracks and porosities in the materials. 
We measured these values to show that their quantity is not significantly great; this is shown by the porosity table. However, it can be seen from the results that a small amount of porosities can significantly impact the hysteresis results. Ideally, the value of the porosity is less than 0.01\%, while there is an inverse relationship between cumulative crack length and porosity. 

- line 217 – circumstances must be replaced with another word.
Corrected.

Reviewer 2 Report

Comments and Suggestions for Authors

This is an interesting work for me. It attempted to  examine the effect of the manufacturing defects on the magnetic hysteresis material. The introduction provides the necessary background information and engineering motivation and the results are also appropriate.  Please see the comments as follows:

1. In section 2.1, the authors stated that the mean particle size was given 33 μm, while most of the particles (95%) were characterized as greater than 20 μm and smaller than 55 μm particle size. Please give a detailed explanation.

2. In section 3.1 and figure 4, Cracks of this type are formed not due to inadequate fusion but are typically caused by the relaxation of frozen internal stresses in this manner. How to get this finding? 

3. In figure 6, the CT image that sample #10 contains a significantly smaller number of cracks and porosity than sample #1 and sample #6. The difference between samples should be in detail introduced, and explain why this happens. 

4. Section 3.2 the main results on the uncertainties in the magnetic hysteresis measurement should be elaborated and discussed. 

5. The main scientific merits should be highlighted in Abstract and Conclusion. 

Author Response

Dear Reviewer,

 Thank you for your work and your comments, which helped us to improve our paper. You can find our answers below, it is highlighted by blue color.

This is an interesting work for me. It attempted to  examine the effect of the manufacturing defects on the magnetic hysteresis material. The introduction provides the necessary background information and engineering motivation and the results are also appropriate.  Please see the comments as follows:

1. In section 2.1, the authors stated that the mean particle size was given 33 μm, while most of the particles (95%) were characterized as greater than 20 μm and smaller than 55 μm particle size. Please give a detailed explanation.

Thank you for the question. Figure 2 shows the particle size distribution, which made by a laser diffraction method to check the distribution of the different samples, this distribution with mean value 33 μm particle size corresponds to the other GB/T 19077-2016 and ISO 13320-2020 standards, knowing this information, our experiments can be repeatable by other researchers.

2. In section 3.1 and figure 4, Cracks of this type are formed not due to inadequate fusion but are typically caused by the relaxation of frozen internal stresses in this manner. How to get this finding?

Thank you for your question. Numerous studies have investigated the causes, locations, orientations, and prevention of porosity and crack formation (Syed, et al. An experimental study of residual stress and direction-dependence of fatigue crack growth behaviour in as-built and stress-relieved selective-laser-melted Ti6Al4V. Materials Science and Engineering). In most cases, cracks parallel to the Z-axis (build direction) are caused by the combination of a high-temperature gradient perpendicular to the build direction and porosities (keyholes) that form between layers (somewhere in the meltpool). The temperature gradient is responsible for the internal stress, and the porosities act as stress concentration points where cracks can start or end.

3. In figure 6, the CT image that sample #10 contains a significantly smaller number of cracks and porosity than sample #1 and sample #6. The difference between samples should be in detail introduced, and explain why this happens. 

The following explanation is added to section 3.1:

The difference between these samples is caused by many small uncertainties during the manufacturing of the samples. These uncertainties can come from different sources, for example, as previously seen, the size of the particles is not identical, there can be some local differences in the material composition between the different samples, the place of the sample in the table, or the precision of the laser can have an effect on the manufacturing process.

4. Section 3.2 the main results on the uncertainties in the magnetic hysteresis measurement should be elaborated and discussed. 

This section aims to evaluate the uncertainty in our measurement chain and the sensitivity of the measured parameters through repetitive measurements.

We extended our discussion in the following way:

The measured data shows that Br is the most sensitive to the measurements, with a relative value of about 1%, while Hc is the most stable of the three selected characteristics, with a sensitivity of less than 0.2%. These numbers characterize the uncertainty of our measurements, all of the changes in the hysteresis curve parameters which are higher than the above-mentioned numbers caused by the thermal and frequency dependency of the examined samples. 

5. The main scientific merits should be highlighted in Abstract and Conclusion. 

The abstract of the paper changed in the following way to highlight better the main scientific merits of the paper:

Additive manufacturing of soft magnetic materials is a promising technology for creating topologically optimized electrical machines. High-performance electrical machines can be made from high silicon content FeSi alloys. FeSI 6.5% material has exceptional magnetic properties; however, manufacturing this steel with the classical cold rolling methodology is not possible due to the brittleness of this material. Laser powder bed fusion technology (L-PBF) offers a solution to this problem. Finding the optimal printing parameters is a challenging task. Nevertheless, it is crucial to resolve the brittleness of the created materials so they can be used in commercial applications.
The temperature dependence of magnetic hysteresis properties of FeSI 6.5% materials are presented in this paper. The magnetic hysteresis properties were examined from 20C to 120C.The hysteresis measurements were made by a precision current generator-based hysteresis measurement tool, which uses fast Fourier transformation-based filtering techniques to increase the accuracy of the measurements. The details of the applied scalar hysteresis sensor and the measurement uncertainties were discussed first in the paper; then, three characteristic points of the static hysteresis curve of the  L-PBF manufactured ten identical toroidal cores were investigated and compared at different temperatures. 
These measurements show that despite the volumetric ratio of the porosities being below 0.5%, the mean crack length in the samples is not significant for the examined samples. These small defects can cause a significant 5% decrement in some characteristic values of the examined hysteresis curve.

Round 2

Reviewer 2 Report

Comments and Suggestions for Authors

The revised manuscript and comments' response can be accepted now.